# Anti-Proliferation and DNA Cleavage Activities of Copper(II) Complexes of N₃O Tripodal Polyamine Ligands

**Doti Serre [1], Sule Erbek [2], Nathalie Berthet [1], Christian Philouze [1], Xavier Ronot [2], Véronique Martel-Frachet [2] and Fabrice Thomas [1,\***

[1] Département de Chimie Moléculaire—UMR CNRS-5250, Université Grenoble Alpes, 38041 Grenoble, France; nathalie.berthet@univ-grenoble-alpes.fr (N.B.)

[2] Institute for Advanced Biology—INSERM-UGA U823, Site Santé, 38700 La Tronche, France; veronique.frachet@univ-grenoble-alpes.fr (V.M.-F.)

\* Correspondence: fabrice.thomas@univ-grenoble-alpes.fr

**Abstract:** Four ligands based on the 2-*tert*-butyl-4-X-6-{Bis[(6-methoxy-pyridin-2-ylmethyl)-amino]-methyl}-phenol unit are synthesized: X = CHO (HL$^{CHO}$), putrescine-pyrene (HL$^{pyr}$), putrescine (HL$^{amine}$), and 2-*tert*-butyl-4-putrescine-6-{Bis[(6-methoxy-pyridin-2-ylmethyl)-amino]-methyl}-phenol (H₂L$^{bis}$). Complexes **1**, **2**, **3,** and **4** are formed upon chelation to copper(II). The crystal structure of complex **1** shows a square pyramidal copper center with a very weakly bound methoxypridine moiety in the apical position. The pKa of the phenol moiety is determined spectrophotometrically at 2.82–4.39. All the complexes show a metal-centered reduction in their CV at E$_p$$^{c,red}$ = −0.45 to −0.5 V vs. SCE. The copper complexes are efficient nucleases towards the $\phi$X174 DNA plasmid in the presence of ascorbate. The corresponding IC$_{50}$ value reaches 7 µM for **2**, with a nuclease activity that follows the trend: **2 > 3 > 1**. Strand scission is promoted by the hydroxyl radical. The cytotoxicity is evaluated on bladder cancer cell lines sensitive (RT112) or resistant to cisplatin (RT112 CP). The IC$_{50}$ of the most active complexes (**2** and **4**) is 1.2 and 1.0 µM, respectively, for the RT112 CP line, which is much lower than cisplatin (23.8 µM).

**Keywords:** copper; tripodal ligand; nuclease; DNA; phenol

## 1. Introduction

Nucleases are important tools for manipulating genes and designing new chemotherapeutic agents [1]. Inorganic complexes are particularly well-suited for this purpose due to the intrinsic properties of the metal ion. Its acidity facilitates the activation of substrates, and hence favors hydrolytic processes [2,3]. On the other hand, redox-active metals may reduce molecular dioxygen into reactive oxygen species (ROS: O₂$^{2−}$, O₂$^{−}$ and OH), which subsequently cleave DNA in an oxidative pathway [2,3].

Numbers of artificial nucleases based on biologically relevant metal ions (manganese, copper, iron) have been developed [3,4]. Assuming that biological metals are less toxic than non-biological ones, these complexes may be used as therapeutic agents, in particular as an alternative to cisplatin in chemotherapy [4,5]. Amongst the biometals, copper has emerged as an attractive target for designing nucleases having an anti-cancer activity [4–10]. Indeed, copper is an essential cofactor for tumor angiogenesis [11], while correlations exist between the copper status and both malignant progression [12,13] and response to therapy in some human cancers [14]. Furthermore, copper(II) is capable of favoring nucleophilic attack at the DNA phosphates, while copper(I) reacts with dioxygen, affording ROS that is potent for strand scission in vitro [15].

The ligand platform plays a pivotal role in modulating the properties of the metal and its interaction (mode and strength) with DNA. Polypyridyl ligands have been widely used for designing copper nucleases because pyridines can intercalate into DNA and give rise

to $Cu^{II}/Cu^{I}$ redox potentials adequate for oxidative cleavage when engaged in coordination [16,17]. They are mostly based on bipyridine [18–20], phenanthroline [18,19,21–27], terpyridine [28], phenanthrene [1,29], and tripodal structures [17,30–33]. These moieties can be further engineered for adding functionalities [34,35]. On the other hand, phenol(ate) donors, especially sterically hindered ones, can be considered as phenoxyl radical precursors, which are efficient moieties for H-atom abstraction [36–38]. When combined with tripodal structures, these radicals may form and be involved in DNA strand scission through hydrogen atom transfer [39–42].

We previously designed copper nucleases from $N_3O$ tripodal ligands featuring one sterically hindered phenol and two pyridines. It is striking to note that the nuclease activity is greatly improved when one pyridine has a methyl substituent in the α position [30]. This group is both electron-donating and sterically demanding. Although these properties are key points for complex activity, the relative influence of the two on nuclease activity is currently unknown. We have further demonstrated that both nuclease activity and anti-proliferative activity against bladder tumor cell lines can be enhanced by the addition of a putrescine chain and a pyrene moiety [43,44]. This functionalization pattern increases the affinity for DNA by both ionic interactions between the putrescine chain and the negatively charged phosphates, and intercalation of the pyrene moiety between consecutive base pairs. These encouraging results prompted us to investigate the role of the pyridine α-substituent in both the nuclease and anti-proliferative activities. We herein describe a new series of ligands derived from HL (Figure 1), in which the methyl substituent of the pyridine is replaced by a highly electron donating methoxy group. With this substitution pattern, we aim to increase the electron density at the metal, while decreasing the steric hindrance, and thus plan to decipher their respective importance in biological activity. This work is also a prelude to the insertion of PEG units on pyridine to further improve drug pharmacokinetics and bioavailability.

**Figure 1.** Nomenclature of the ligands under their neutral amino forms.

We present the synthesis of the ligands $HL^{CHO}$, $HL^{Pyr}$, $HL^{amine}$, and $H_2L^{bis}$ (Figure 1), as well as the solution chemistry and spectroscopy of their copper complexes (**1**, **2**, **3**, and **4**, respectively). We investigated their nuclease activity and anti-proliferative activity towards bladder tumor cells.

## 2. Results and Discussion

### 2.1. Synthesis and Structure of the Complexes

The preparation of the series of ligands starts with the synthesis of the aldehyde precursor HL$^{CHO}$ (Figure 1), which is next engaged in a reductive amination [43]. When $N^1$-(pyren-1-ylmethyl)butane-1,4-diamine was reacted with HL$^{CHO}$, the ligand HL$^{pyr}$ was obtained. By using the *tert*-butyl-N-(4-aminobutyl)carbamate instead of $N^1$-(pyren-1-ylmethyl)butane-1,4-diamine, followed by acid deprotection, the ligand HL$^{amine}$ was obtained instead. Finally, the reaction of neutral HL$^{amine}$ with one molar equivalent of HL$^{CHO}$ affords the binucleating ligands H$_2$L$^{bis}$ (Figure 1). The mononuclear copper-phenolate complexes **1–3** were prepared in situ by mixing stoichiometric amounts of CuCl$_2$ 2H$_2$O, NEt$_3$, and the appropriate ligand in DMF (Scheme 1); the binuclear copper-phenolate complex **4** was prepared similarly by using two molar equivalents of metal salt instead of a single one. Both the nuclearity and nature of the complexes was established by mass spectrometry, UV-Vis, and EPR spectroscopies (see below). The mass spectra of the complexes are depicted in Figure S9. The spectrum of **1** shows a main peak corresponding to a monocation at $m/z = 481.1421$. It shows the expected pattern for a copper containing complex and corresponds to [**1**]$^+$ (calculated $m/z = 481.1410$). The pyrene-appended complex **2** exhibits a main peak at $m/z = 384.1660$ (dication), which is assigned to [**2** + H]$^{2+}$, while complex **3** shows a main peak at $m/z = 277.1268$ assigned to [**3** + H]$^{2+}$. For comparison, complex **4** demonstrates a peak at $m/z = 1020.4$, corresponding to [**4** + 2H]$^+$, without evidence for a monometalated species. The (2:1) (M:L) stoichiometry of **4** was further confirmed by the Jobs method (Figure S13).

HL$^R$ + CuCl$_2$ $\xrightarrow[\substack{\text{H}_2\text{O:DMF} \\ \text{90:10}}]{\text{pH = 7}}$ **1** (R = CHO)
**2** (R = Pyr)
**3** (R = amine)

H$_2$L$^{bis}$ + 2 CuCl$_2$ $\xrightarrow[\substack{\text{H}_2\text{O:DMF} \\ \text{90:10}}]{\text{pH = 7}}$ **4**

**Scheme 1.** Nomenclature of the complexes and their schematized synthesis. Complex **1** was isolated as single crystals; **2–4** were generated in situ for physical and biological characterizations.

The structure of **1** was substantiated by X-Ray diffraction (Figure 2). The other complexes proved to be difficult to isolate at the solid state, presumably because of the hygroscopic nature of the compounds (due to the presence of secondary amines) and the large flexibility of the putrescine arm. The structure of **1** shows a square pyramidal copper center where the tertiary amine nitrogen N3, the pyridine nitrogen N2, the phenolate oxygen O1, and the chloride Cl coordinate in the basal plane. The methoxypyridine is positioned apically, but this group is almost uncoordinated (Cu-N1 bond distance of 2.773(5) Å). This behavior is in sharp contrast with the methylpyridine analog of $^{Me}$**1** that binds apically at a much shorter Cu-N1 bond distance of 2.303(2) Å [44]. The Cu-N3, Cu-N2, Cu-O1, and Cu-Cl bond distances are 2.034(3), 2.017(3), 1.902(3), and 2.251(1) Å, respectively. They are significantly shorter than in $^{Me}$**1**, as a result of the lengthening of the axial Cu-N1 bond. The angle between the pyridine (equatorial) and substituted pyridine (axial) rings differs significantly between $^{Me}$**1** and **1**; although 70° in $^{Me}$**1**, it does not exceed 36° in the case of **1**. Furthermore, the distance between the centroids of these two rings reaches 4.21 Å for $^{Me}$**1**, while it is only 3.80 Å for **1**. These structural features indicate that the more electron-rich methoxypyridine ring has a higher propensity to stack on the equatorial pyridine, weakening axial coordination.

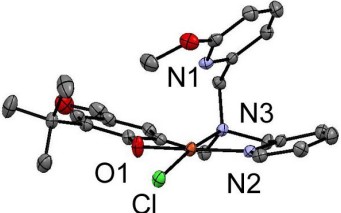

**Figure 2.** X-Ray crystal structure of **1**. The hydrogen atoms are omitted for clarity.

## 2.2. Spectroscopic Characterization

The complexes were formed in situ by mixing equimolar amounts of copper(II) chloride and the ligand (**1**, **2**, **3**) or two molar equivalents of the copper salt with the ligand (**4**). It is worth noting that a solution of single crystals of **1** shows the same spectroscopic signature than the in situ generated complex.

The EPR spectra of **1**–**4** show a clean axial ($S = \frac{1}{2}$) signal at pH 7 (Figures S11 and S12), disclosing the formation of a unique species in every case. Importantly, the intensity of the signal is identical for **1**–**3**. On the basis of the similarity of the donor set between HL, HL$^{Pyr}$, and HL$^{amine}$ and given the resolution of the crystal structure of **1** (see above), we assume that the complexes formed in all three cases are mononuclear complexes. The spectrum of **4** is similar to that of **3**, although its intensity is twice as large. The magnetic interaction between the two copper centers is therefore negligible, which is not surprising given the length and flexibility of the linker connecting the tripodal units. Note that the shape of the spectra is similar for **2**–**4**, consistent with an identical N$_3$O coordination sphere, and combined with a similar electron-donating effect of the phenolic para-substituent.

The experimental spectra at arbitrary chosen pH values, which correspond to pKa + 1 and pKa − 1 (see below) where the acidobasic function is the phenolic hydroxyl, together with simulations are depicted in Figures 3 and S13–S15. A general trend is that g$_\perp$ is smaller than g$_{//}$, as expected for a d$_{x2\text{-}dy2}$ ground spin state. This ground state is consistent with the square pyramidal geometry of the copper(II) ion in the solid state structure. The spin Hamiltonian parameters differ significantly between pH = pKa−1 (1.80–3.34, for **1**–**4** see Table 1) and pH = pKa + 1 (3.80–5.35, for **1**–**4** see Table 1) for a given complex in the non–buffered medium. This suggests a pH-induced change in metal ion geometry and thus the existence of an acidobasic equilibrium involving a coordinating moiety, which we have attributed to the phenol/phenolate couple. On the other hand, the g$_{//}$/A$_{//}$ ratio has previously been correlated with copper ion distortion [45]. The g$_{//}$/A$_{//}$ ratio is 128 cm at low pH (phenol complex), corresponding to a main square planar geometry (very weak apical coordination), whereas it is 132–136 cm at high pH (phenolate). The copper environment is therefore more distorted in deprotonated complexes, which is likely a consequence of the stronger binding of the deprotonated phenolic moiety. It is worth noting that the EPR parameters diverge slightly between pH = pKa + 1 (unbuffered) and pH = 7 (buffered), which can be attributed to the binding of a different exogenous ion (chloride or water molecule depending on NaCl concentration.

In order to confirm the nature of the acidobasic residue and determine precisely the pKa, we conducted UV-Vis titrations of the four complexes over the pH range 2–7. The spectral evolution of **2** is depicted in Figure 4, while the other ones are shown in Figures S16–S18. Since the evolutions are similar, we will comment only on the spectra of **2**. Thus, the spectrum at pH = 6.94 is dominated by an intense band at 462 nm (Table 2), which is assigned to a phenolate-to-copper charge transfer (LMCT) transition (see TD-DFT section). As the pH decreases, the intensity of the LMCT decreases, indicating the protonation of the phenolate. At pH = 2.55, the 470 nm band has disappeared, while a low intensity band is still observed at 680 nm, corresponding to copper(II) d-d transitions.

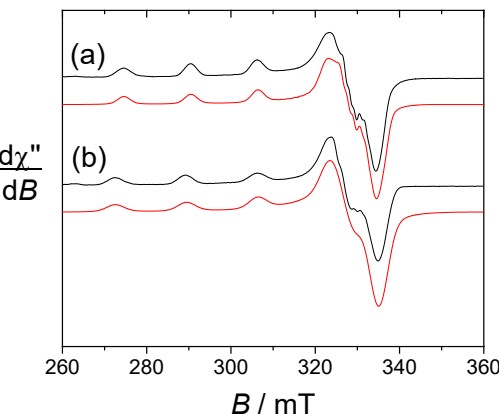

**Figure 3.** X-Band EPR spectra of 0.5 mM solutions of **2** in a (water:DMF) (90:10) medium at (**a**) pH = 5.15 and (**b**) pH = 3.11. Black: Experimental spectrum; Red: spectral simulation using the EASYSPIN 5.2.35 software and the parameters given in Table 1. [NaCl] = 0.1 M; *T*, 100 K; microwave frequency, 9.44 GHz; microwave power, 4 mW; mod. Freq., 100 KHz; mod. Amp. 0.4 mT.

**Table 1.** Spin Hamiltonian parameters for the copper complexes.

| Complex | pH | $g_\perp$ | $g_{//}$ | $A_\perp$ | $A_{//}$ | g/A [c] |
|---------|-----|-------|-------|-------|-------|---------|
| 1 | 1.80 [a] | 2.056 | 2.266 | 1.7 | 16.9 | 128 |
| 2 | 3.11 [a] | 2.056 | 2.266 | 1.7 | 16.9 | 128 |
| 3 | 3.25 [a] | 2.055 | 2.266 | 1.4 | 16.9 | 128 |
| 4 | 3.34 [a] | 2.055 | 2.266 | 1.4 | 16.9 | 128 |
| 1 | 3.80 [a] | 2.053 | 2.264 | 1.4 | 16.4 | 132 |
| 2 | 5.15 [a] | 2.053 | 2.261 | 1.4 | 15.9 | 136 |
| 3 | 5.26 [a] | 2.053 | 2.261 | 1.4 | 15.9 | 136 |
| 4 | 5.35 [a] | 2.054 | 2.262 | 1.4 | 16.0 | 135 |
| 1 | 7 [b] | 2.056 | 2.257 | 1.4 | 16.3 | 132 |
| 2 | 7 [b] | 2.054 | 2.254 | 1.5 | 16.6 | 130 |
| 3 | 7 [b] | 2.054 | 2.254 | 1.5 | 16.6 | 130 |
| 4 | 7 [b] | 2.054 | 2.254 | 1.5 | 16.6 | 130 |

[a] In ($H_2O$:DMF) (90:10) frozen solution containing 0.1 M NaCl. $HClO_4$ was added to adjust the pH values to ca. pKa-1 and pKa + 1. [b] In ($H_2O$:DMF) (90:10) frozen solution. [Tris−HCl] = 0.05 M, [NaCl] = 0.02 M. [c] ratio expressed in cm.

**Table 2.** pKa values and electronic spectra of the copper complexes [a].

| | | Phenolate Form | | | | Phenol Form [c] | |
|---------|-----|----------------|-----|---------|-----|-----------------|-----|
| | | Charge Transfer | | d-d Band | | d-d Band | |
| Complex | pKa | $\lambda_{max}$ | $\varepsilon$ | $\lambda_{max}$ | $\varepsilon$ | $\lambda_{max}$ | $\varepsilon$ |
| 1 | $2.82 \pm 0.02$ | 482 | 662 [b] | 680 | 170 | 680 [c] | <90 [c] |
| 2 | $4.15 \pm 0.01$ | 462 | 1040 | 670 | 101 | 680 | 66 |
| 3 | $4.25 \pm 0.01$ | 461 | 780 | 670 | 232 | 687 | 147 |
| 4 | $8.77 \pm 0.03$ [d] | 457 | 2005 | 670 | 328 | 687 | 147 |

[a] Fitted using the SPECFIT 3.0.37 software (from Biologic, Seyssinet-Pariset, France). The spectra were recorded in a ($H_2O$:DMF) (90:10) solution containing [NaCl] (0.1 M). $HClO_4$ or NaOH was added to adjust the pH. [b] Phenolate-to-copper charge transfer transition. [c] At the lowest pH investigated (1.8) the phenolate-to-copper charge transfer was still observable. [d] The value indicated corresponds to the -log$\beta_2$ value associated to the equilibrium: $[(H_2L^{bis})Cu_2]^{4+} \rightleftharpoons 2H^+ + [(L^{bis})Cu_2]^{2+}$.

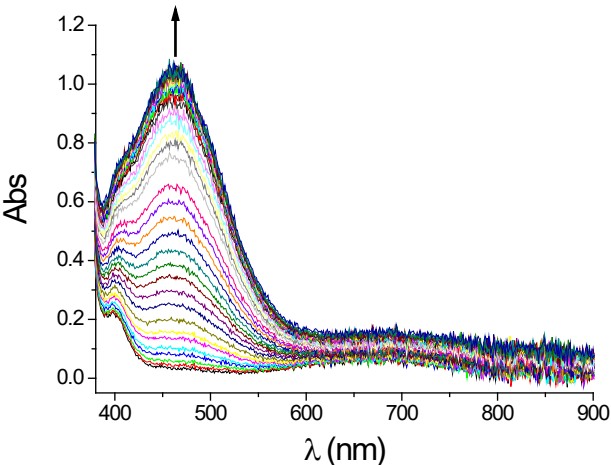

**Figure 4.** Electronic spectrum of **2** as a function of the pH. 1 mM (H$_2$O:DMF) (90:10) solution containing 0.1 M [NaCl], T = 298 K, *l* = cm. The pH is varied from 2.55 to 6.94, the arrow indicates spectral changes upon addition of base.

The phenol pKas were determined at 2.82 $\pm$ 0.02, 4.15 $\pm$ 0.01, 4.25 $\pm$ 0.01, and 8.77 $\pm$ 0.03 (Log$\beta$ for a two-proton process in the latter case) for **1**, **2**, **3**, and **4**, respectively. The lower value of **1** is explained by the electron-withdrawing properties of the aldehyde o-substituent. It is worth noting that the pKa values of **2**, **3**, and **4** (referred to one proton in the latter case, e.g., 4.38) are in a narrow range, consistent with a similar electronic effect of the phenol para substituent (same alkyl ammonium chain). It should be emphasized that these pKa values are 0.22 to 0.48 units higher than for analogous methyl complexes [44]. We interpret this increase in pKa by a decrease in Lewis acidity at the metal center due to the greater electron-donating capacity of the methoxy-substituted pyridine. This trend validates our working hypothesis that a change in the $\alpha$-substituent of the pyridine would influence the electron density at the metal center. Importantly, all pKa values are much lower than those reported for complexes with axially bound phenolates [46,47], supporting an equatorial positioning of the phenolate in **1**–**4**. This interpretation is corroborated by the crystal structure of **1**.

*2.3. Electrochemistry*

The electrochemical behavior of the complexes has been investigated by cyclic voltammetry (CV) in a (H$_2$O:DMF) (90:10) solution containing 0.1 M NaCl as supporting electrolyte (Figures 5, S19 and S20). Redox potentials are summarized in Table 3. The CV curve for **1** demonstrates a cathodic peak at $E_p^{c,red}$ = −0.45 V, which is associated with an oxidation peak at $E_p^{a,red}$ = −0.14 V (red curve in Figure 5). This pair of semi-reversible peaks ($\Delta E$ = 0.31 V) is assigned to the Cu$^{II}$/Cu$^{I}$ redox couple, in which significant structural rearrangements are associated with electron transfer. The apparent $E_{1/2}$ calculated as ($E_p^{c,red}$ + $E_p^{a,red}$)/2 is −0.30 V. An irreversible oxidation is also observed at $E_p^{a,ox}$ = +0.96 V and attributed to phenolate oxidation. The Cu$^{II}$/Cu$^{I}$ redox couple in **2**, **3**, and **4** is again observed as a pair of separated peaks, similar to **1**. For **2**, the potentials values are $E_p^{c,red}$ = −0.51 V, $E_p^{a,red}$ = +0.16 V ($\Delta E$ = 0.67 V, apparent $E_{1/2}$ = −0.17 V), while for **4** they are $E_p^{c,red}$ = −0.47 V, $E_p^{a,red}$ = +0.07 V ($\Delta E$ = 0.54 V, apparent $E_{1/2}$ = −0.20 V). A broadening of the cathodic peak ($E_p^{c,red}$) is observed for **3**, presumably due to the presence of primary amines. This precludes a thorough comparison between **3** and the other complexes. The overall increase in $\Delta E$ in **2** and **4** compared with **1** reflects slower electron transfer, likely associated with greater structural rearrangements for the complexes appended by the alkylammonium chain. In addition, the significant anodic shift of $E_p^{a,red}$ (in comparison to **1**) together with small variations in $E_p^{c,red}$ suggest enhanced stability of the reduced form. Further comparison with previously reported methyl derivatives is unfortunately hampered by the broadening of the redox waves. Finally, two oxidation

peaks are observed for **2**, **3**, and **4**. These are attributed to sequential oxidations of the phenolate unit [30,43,44,47–51] and an amine of the polyamine chain [43,44].

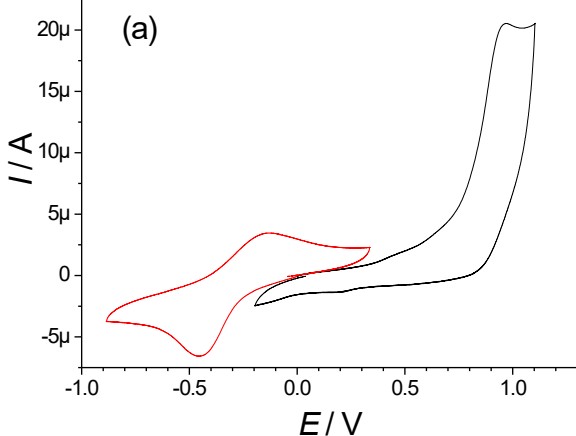

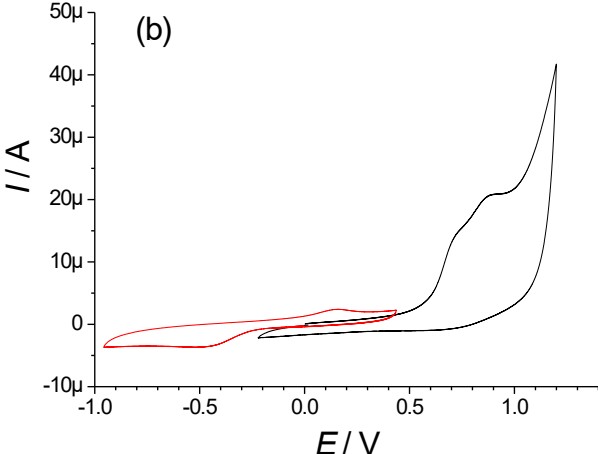

**Figure 5.** Cyclic voltammetry curves of (**a**) **1** and (**b**) **2** in a water:DMF 90:10 medium at pH = 7. [Tris−HCl] = 0.05 M, [NaCl] = 0.02 M; T, 298 K; scan rate, 0.1 V s$^{-1}$; The potentials are quoted relative to the SCE reference electrode. Red: scan in reduction; Black: scan in oxidation.

**Table 3.** Electrochemical data of the complexes at pH 7 [a].

| Complex | $E_p^{c,red}$ | $E_p^{a,red}$ | $E_p^{a,ox}$ |
|---|---|---|---|
| **1** | −0.45 | −0.14 | 0.96 |
| **2** | −0.51 | +0.16 | 0.73, 0.88 |
| **3** | −0.45 | - [b] | 0.69, 0.9 [b] |
| **4** | −0.47 | +0.07 | 0.56, 0.73, 0.88 |
| PyrPt [c] | - | - | 0.86, 1.0 [b] |

[a] In a (H$_2$O:DMF) (90:10) solution, [Tris−HCl] = 0.05 M, [NaCl] = 0.02 M. Reference, SCE; T, 298 K; Br: broad. [b] Ill-defined. [c] PyrPt: N$^1$-(pyren-1-ylmethyl)butane-1,4-diamine.

## 2.4. DFT and TD-DFT Calculations

In order to gain insight on the structures of the complexes and their UV-Vis signatures, we performed DFT calculations (Figure 6). In order to minimize computation time, we considered a model in which the para substituent of the phenol is a methyl group. We first optimized the structure of this model in its phenolate form with a chloride ligand. The methoxy group was arbitrary orientated either towards the N of the pyridine or away

from the N. Strikingly, when orientated towards the N it creates significant steric hindrance that weakens the coordination of the methoxy pyridine (Cu-N distance of 2.80 Å). The almost uncoordinated methoxypyridine instead stacks up on the equatorial, unsubstituted pyridine. When the methoxy group is orientated away from the N, the steric hindrance is lower, allowing coordination of the methoxy pyridine (Cu-N bond distance of 2.29 Å), and no further stacking is observed. The most stable conformation is the first (by 3 kcal/mol), in agreement with the solid-state structure of **1** (with a chloride ligand). It is worth noting that a second conformation has been identified (1 kcal/mol less stable) in which the tertiary amine occupies the axial position. When the chloride ligand is replaced by a water molecule, a different behavior is observed, in which the methoxy oxygen establishes a H-bond with a hydrogen of the coordinated water molecule. As a result, the conformation in which the methoxy group points away from the pyridine nitrogen is favored. We have also optimized the structure of the phenol complexes, in which this latter group is located apically due to its lower coordinating capacity. Both orientations of the methoxy group lead to isoenergetic structures when the exogenous ligand is water, but the structure in which the methoxy points away from the N is more stabilized for the chloro complex (by 4.8 kcal/mol). For the lowest energy conformations (phenol and phenolate, chloro, and water adducts), we calculated the UV-Vis spectra by TD-DFT calculations. For the phenolate complexes, the calculations predict a relatively intense phenolate-to-copper charge transfer in the visible region, and the agreement between theory and experiment is excellent when the water adduct is taken in account (Table 4). For the phenol complex, a low-intensity d-d band is predicted in the low-energy range (Table 4), and once again agreement between theory and experiment is excellent when the water adduct is considered.

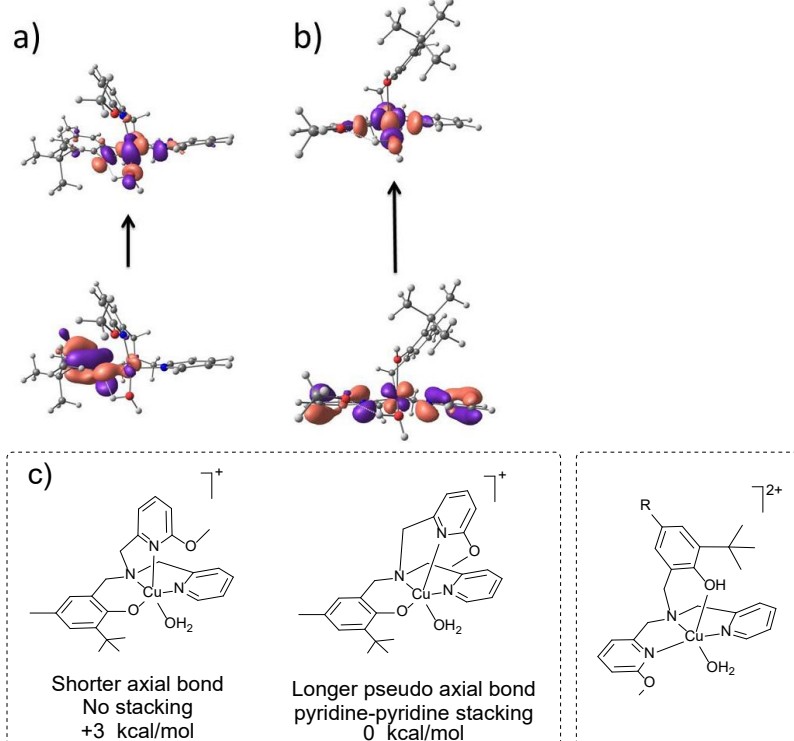

**Figure 6.** TD-DFT assignment of the main electronic excitations in the (**a**) phenolate (βHOMO → βLUMO) and (**b**) phenol (βHOMO-5 → βLUMO) complexes where the exogenous ligand is a water molecule. (**c**) Structures of the model complexes used for the calculations (B3LYP/6–31g*/SCRF, contour value: 0.040).

**Table 4.** TD-DFT assignment of the main visible band [a].

| Protonation State | Exogenous Ligand | Main Contribution to the Electronic Excitation | $\lambda_{cald}$ [nm] ($f$) |
|---|---|---|---|
| Phenolate | chloride | $\beta$HOMO $\rightarrow$ $\beta$LUMO (LMCT) | 510 (0.0376) |
| Phenolate | water | $\beta$HOMO $\rightarrow$ $\beta$LUMO (LMCT) | 563 (0.0646) |
| Phenol | chloride | $\beta$HOMO-7 $\rightarrow$ $\beta$LUMO (LMCT) | 855 (0.0031) |
| Phenol | water | $\beta$HOMO-5 $\rightarrow$ $\beta$LUMO (dd) | 661 (0.0012) |

[a] B3LYP/6–31g*/SCRF (water). Most stable conformation for the phenolate complex.

### 2.5. DNA Cleavage

We investigated the nuclease activity of the complexes towards plasmidic supercoiled DNA. The reaction was monitored by gel electrophoresis in a medium constituted by a mixture (water:DMF) (90:10) in a phosphate buffer (10 mM) at pH = 7.2. Cleavage of the closed circular supercoiled (SC) plasmidic DNA affords either the nicked circular (NC) form (single-strand breakage) or the linear (L) form (double-strand breakage).

The DNA cleavage activity was first investigated in the absence of an exogenous agent (Figure 7, Table 5). Complex **1** cleaved 20% of DNA (single-strand cleavage) at the highest concentration investigated (50 μM, Figure 7a), and proved to be a better nuclease than its methyl congener (no cleavage at this concentration) [44]. Complex **2** and **4** (Figure 7b,d) behave drastically differently since the gel electrophoresis pattern shows the disappearance of the SC form as DNA concentration increases, with no appearance of the NC or L form. This behavior has already been observed for the methyl derivative as well as for polyamines [44,52]. It results from DNA condensation, which is favored by neutralization of the phosphate by protonated polyamines. This hypothesis is further supported by the observation of a delay in the migration of the SC form before its total disappearance. Under these conditions, it is not possible to conclude about the nuclease activity of **2** and **4**. Complex **3** behaves in a more "classical" manner, in that the NC form was visible in addition to the SC form. The NC/SC ratio is clearly in favor of the NC form (single-strand cleavage) on the gel electrophoresis at 50 μM complex concentration, reflecting significant cleavage activity. Since aggregation can occur, nuclease activity could be overestimated, so that only an upper limit of 65% cleavage can be calculated. However, complex **3** appears visually to be a much better nuclease than complex **1**, due to the incorporation of a putrescine chain.

**Table 5.** DNA cleavage activity at pH = 7.2 [a].

| Complex | No Exogenous Agent | Mercaptan | Ascorbic Acid |
|---|---|---|---|
| **1** | >50 | 40 | 30 |
| **2** | - [b] | 13 | 7 |
| **3** | >40 | 22.5 | 16 |
| **4** | - [b] | - [b] | >18 |

[a] Expressed as the concentration of complex (in μM) that produces 50% of cleavage of $\phi$X174 supercoiled DNA. [DNA] = 20 mM base pairs; T = 37 °C; t = 1 h; [reductant] = 0.8 mM; phosphate buffer 10 mM; pH = 7.2; water:DMF (90:10). [b] The gel electrophoresis displays only the disappearance of the SC form, due to significant fragmentation or condensation of DNA (see the text). The $IC_{50}$ value cannot be determined under these conditions.

The DNA cleavage activity was further investigated in the presence of two distinct reductants, namely mercaptoethanol and ascorbic acid. As depicted in Figure 8 and Table 5, the nuclease activity is increased when the reductant is present, suggesting a different cleavage mechanism (oxidative versus hydrolytic). Amongst the two reductants, ascorbic

acid gave the best results. Since mercaptoethanol is a potential ligand due to its thiol functions, we will discuss only the results with ascorbic acid in the next section. It is worth noting that the cleavage can be now quantified for **1**, **2**, and **3** since the concentration of complex required to achieve 50% cleavage is lower in the presence of ascorbic acid than without. It is in these cases smaller than that which promotes DNA condensation. For **4**, multiple bands are observed at 15 μM, which disappear at higher concentrations, precluding an accurate determination of IC$_{50}$ (Figure 8d).

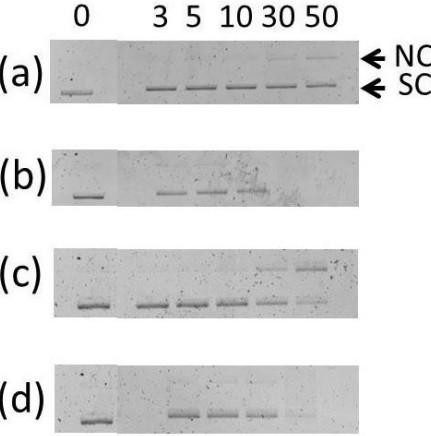

**Figure 7.** Cleavage of the $\phi$X174 supercoiled DNA by the copper complexes. The reaction is monitored by agarose gel electrophoresis in the absence of reductant at 37 °C for 1 h. [DNA] = 20 mM (base pairs); (water:DMF) (90:10) mixture; phosphate buffer (10 mM);pH = 7.2. Abbreviations: NC: nicked circular, SC: supercoiled. The concentrations of the complexes are expressed in μM. (**a**) **1**; (**b**) **2**; (**c**) **3** and (**d**) **4**.

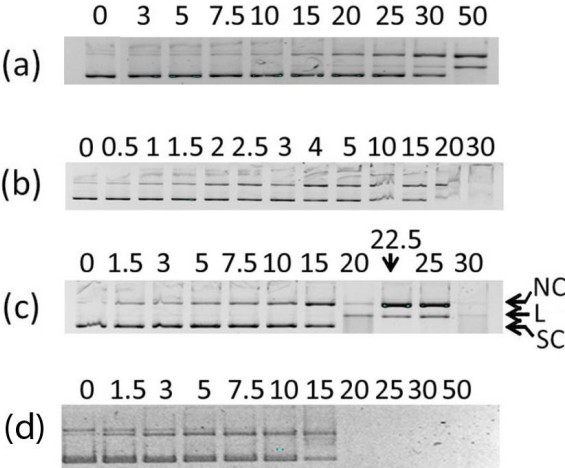

**Figure 8.** Cleavage of the $\phi$X174 supercoiled DNA by the copper complexes. The reaction is monitored by agarose gel electrophoresis in the presence of reductant at 37 °C for 1 h. [DNA] = 20 mM (base pairs); (water:DMF) (90:10) mixture; phosphate buffer (10 mM); [ascorbate] = 0.8 mM; pH = 7.2. Abbreviations: NC: nicked circular, SC: supercoiled, L: linear. The concentrations of the complexes are expressed in μM. (**a**) **1**; (**b**) **2**; (**c**) **3**; (**d**) **4**.

The most efficient nuclease is complex **2**, which cleaves 50% DNA at the concentration of 7 μM (Figure 8b). Complex **3**, which does not feature the pyrene moiety, is less active (16 μM for 50% of cleavage, Figure 8c), while complex **1** which features neither a pyrene, nor a putrescine chain as DNA anchoring function, is the poorest nuclease (30 μM for 50% of cleavage, Figure 8a). Interestingly, complex **3** catalyzes total DNA cleavage at a concentration of 25 μM, through both single and double-strand cleavage: The ratio is 60%

of double-strand cleavage and 40% of single-strand cleavage. Thus, the presence of a free terminal amine is important for the activity.

Important information about the cleavage mechanism was obtained from experiments in the presence of ascorbic acid and several scavengers at complex concentrations inducing significant cleavage. As illustrated in Figure 9 for **1** and **2** (and Figure S22 for **3**), the addition of superoxide dismutase (lane 4), NaN$_3$ (lane 5), or catalase (lane 7) does not induce changes in the cleavage profile, ruling out the involvement of superoxide, hydrogen peroxide, or singlet dioxygen in the reaction. In contrast, DMSO and to a lesser extend ethanol slightly inhibit the nuclease activity, pointing out the involvement of the hydroxyl radical in the reaction. Finally, EDTA, which is a strong copper chelator, has a significant inhibitory effect (lane 6). It can therefore be reasonably assumed that the reaction is initiated by the reduction of copper(II) to copper(I), which next activates dioxygen to form the reactive hydroxyl radical. The hydroxyl radical is the species ultimately involved in DNA strand scission [4,15].

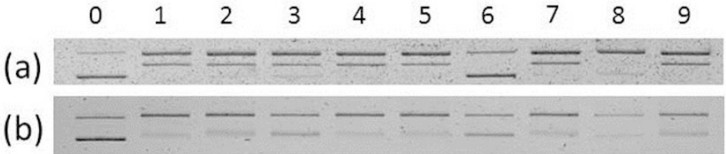

**Figure 9.** Cleavage of the $\phi$X174 supercoiled DNA by the copper complexes. The reaction is monitored by agarose gel electrophoresis in the presence of reductant at 37 °C for 1 h. [DNA] = 20 mM (base pairs); (water:DMF) (90:10) mixture; phosphate buffer (10 mM); [ascorbate] = 0.8 mM; pH = 7.2. (**a**) **1**, 50 µM; (**b**) **2**, 20 µM. Lane 0, DNA control; lane 1, DNA + complex (no scavenger); lanes 2–10, DNA + complex in the presence of various scavengers and agents: lane 2, ethanol; lane 3, DMSO (2 mL); lane 4, superoxide dismutase (0.5 unit); lane 5, NaN$_3$ (100 mM); lane 6, EDTA (10 mM); lane 7, catalase (0.1 unit); lane 8, Hoechst 33258 (100 mM); lane 9, NaCl (350 mM).

*2.6. Mode of Binding of the Best Nuclease 2*

In order to gain insight into the binding of the best nuclease, which is complex **2,** we investigated the nuclease activity in the presence of two binding agents: The minor groove binder Hoechst 33258 (Figure 9, lane 8) and NaCl (Figure 9, lane 9). The latter does not significantly alter the cleavage, but Hoechst 33258 proved to inhibit it significantly. This result suggests a minor groove binding of the putrescine chain. On the other hand, conjugated aromatic rings are known to intercalate into DNA. With the aim of confirming that the pyrene moiety of **2** is intercalated into DNA, we monitored its fluorescence at 468 nm upon titration with DNA. The pyrene units form excimers in solution, which give an intense fluorescence of around 470 nm. When DNA is added to **2**, the fluorescence at 468 nm is progressively quenched, showing that a process interferes with excimer formation, which is attributed to intercalation of the pyrene moiety [53]. The spectral changes were fitted by using a Scatchard–Von Hippel model (Figure 10) [54], giving a K value of $1.6 \times 10^6$ M$^{-1}$ (n = 2). It is within the same order of magnitude than that measured for the methyl derivative [44], disclosing similar interactions with DNA. This further indicates that the main determinant for the interaction is not the nearby environment around the copper center, but ionic interactions mediated by the positively charged copper tripodal unit and the chain, as well as the intercalation of the pyrene group. Noteworthy, the K value is significantly higher than that of ethidium bromide ($4.9 \times 10^5$) [55], confirming the high affinity of **2** for DNA.

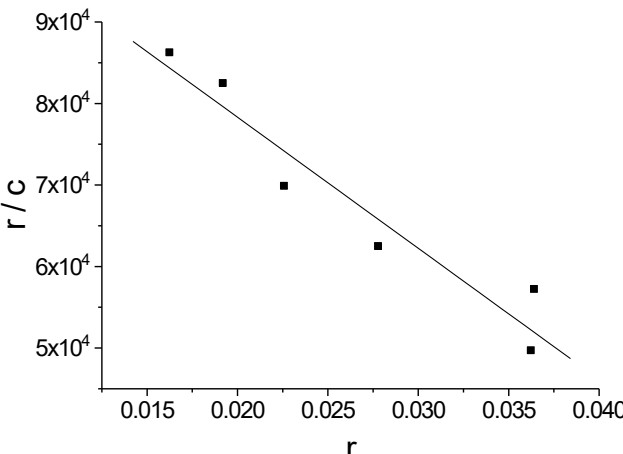

**Figure 10.** Fluorescence data fitted according to a Scatchard–Von Hippel model for the binding of **2** to CT DNA. The medium is a (water: DMF) (90:10) mixture containing 0.02 M NaCl. The pH is adjusted at 7 by a Tris−HCl buffer (0.05 M). T, 298 K.

### 2.7. Anti-Proliferative Activity

We investigated the anti-proliferative activity of complexes **1–4** on two bladder cancer cell lines by MTT assays: The RT112 cell lines are sensitive to cisplatin, whereas the RT112-CP are resistant to cisplatin. The results obtained after 48 h incubation are summarized in Table 6 and compared to those obtained with cisplatin.

**Table 6.** Anti-proliferative activity [a].

| Complex | $IC_{50}$ (μM) | | RF |
|---|---|---|---|
| | **RT112** | **RT112-CP** | |
| Cisplatin | $9.1 \pm 1.2$ | $23.8 \pm 1.2$ | 2.6 |
| **1** | $10.9 \pm 0.2$ | $13.2 \pm 0.2$ | 1.2 |
| **2** | $2.0 \pm 0.2$ | $1.2 \pm 0.1$ | 0.6 |
| **3** | $10.8 \pm 0.2$ | $10.5 \pm 0.9$ | 1.0 |
| **4** | $1.6 \pm 0.2$ | $1.0 \pm 0.1$ | 0.6 |

[a] The concentration resulting in 50% loss of cell viability relative to untreated cells ($IC_{50}$) was determined from dose-response curves. Results represent means $\pm$ SD of three independent experiments. RF is the ratio of $IC_{50}$ against RT112-CP to RT112 cell lines. RT112-CP and RT112 are cisplatin-resistant and sensitive cell lines, respectively.

Complexes **2** and **4** exhibit the highest anti-proliferative activity, with remarkable $IC_{50}$ values of 1–2 μM against both RT112 and cisplatin-resistant cells RT112-CP. The cytotoxicities of **2** and **4** are comparable to those of their methyl derivatives. They are much lower than those of cisplatin, which exhibits $IC_{50}$ values of 9.1± 1.2 and 23.8± 1.2 μM against these two cell lines, respectively. Among the whole tripodal series of antiproliferative agents (incl. both the methylpyridine [44] and methoxypyridine derivatives), the best compounds proved to be **2** and **4** against the RT112-CP line (1.2 $\pm$ 0.1 and 1.0 $\pm$ 0.1 μM). Direct correlation between the nuclease activity and the cytotoxicity yet remains to be established, however, complex **2** proved to be both the best nuclease and the best anti-proliferative complex in the series. Finally, the resistance to cisplatin can be expressed through the resistance factor RF. It is calculated as the ratio of $IC_{50}$ on RT112CP cell lines over $IC_{50}$ on RT112 cell lines, the higher values being indicative of greater resistance. The RF is 2.6 for cisplatin but it is lower than 1.2 for the copper complexes, with remarkable values of 0.6 for complexes **2** and **4**. Hence, all the complexes appended by one putrescine chain overcome the resistance to cisplatin of RT112-CP cells.

## 3. Materials and Methods

### 3.1. Materials and Instruments

All chemicals were of reagent grade and were used without purification. NMR spectra were recorded on a Bruker AM 300 ($^1$H at 300 MHz) spectrometer (Bruker France SAS, Wissembourg, France). Chemical shifts are quoted relative to tetramethylsilane (TMS). Mass spectra were recorded on a Thermofinningen (EI/DCI) or a Bruker Esquire ESI-MS apparatus. For pKa determinations, UV/Vis spectra were recorded on a Cary Varian 50 spectrophotometer equipped with a Hellma immersion probe (1.000 cm path length, Hellma France, Paris, France). The temperature in the cell was controlled using a Lauda M3 circulating bath (Lauda France, Roissy-en-France, France) and the pH was monitored using a Methrom 716 DMS Titrino apparatus (Metrohm France, Villebon Courtaboeuf, France). A least square fit of the titration data was realized with the SPECFIT 3.0.37 software (from Biologic, Seyssinet-Pariset, France). The fluorescence spectra were recorded on a Cary Eclipse spectrometer. X-band EPR spectra were recorded on a Bruker EMX Plus spectrometer equipped with a Bruker nitrogen flow cryostat and a high sensitivity cavity (Bruker France SAS, Wissembourg, France). Spectra were simulated using the Easyspin 5.2.35 software [56]. Electrochemical measurements were carried out using a CHI 620 potentiostat. Experiments were performed in a standard three-electrode cell under argon atmosphere. A glassy carbon disc electrode (3 mm diameter), which was polished with 1 mm diamond paste, was used as the working electrode. The auxiliary electrode is a platinum wire, while a SCE was used as reference.

### 3.2. Synthesis

3-*tert*-butyl-4-hydroxy-benzaldehyde and $N^1$-(pyren-1-ylmethyl)butane-1,4-diamine were prepared according to literature procedures.

(6-methoxy-pyridin-2-ylmethyl)-pyridin-2-ylmethyl-amine.

6-methoxy-pyridin-2-carbaldehyde (2.74 g, 0.02 mol) and Pyridin-2-ylmethylamine (2.16 g, 0.02 mole) were each dissolved in methanol (15 mL). The two solutions were cooled down at 0 °C (ice bath) and mixed together. The ice bath was next removed, and the solution was stirred for 1 h at room temperature. The reaction mixture was cooled down at 0 °C and NaBH$_4$ (1 g, 0.026 mole) was slowly added in small portions (3 h). The solution was further stirred for 12 h at room temperature. Water was then added to the reaction mixture (30 mL), which was neutralized, extracted with CH$_2$Cl$_2$, and dried over anhydrous Na$_2$SO$_4$. To the residue were added CHCl$_3$ and pentane. The solution was cooled at −18 °C overnight. The supernatant was collected and concentrated under vacuum to give 3.66 g of an orange oil (yield: 80%). NMR $^1$H (400 MHz, CDCl$_3$): δ (ppm) = 3.87 (s, 2H); 3.90 (s, 3H, OMe); 6.57 (d, 1H); 6.85 (d, 1H); 7.13 (m, 1H); 7.35 (d, 1H); 7.48 (d, 1H); 7.62 (td, 1H); 8.53 (d, 1H). NMR $^{13}$C (Q.DEPT, 400 MHz, CDCl$_3$): δ (ppm) = 53.41 (CH$_3$, OMe); 54.50; 54.98 (2 CH$_2$); 108.85; 114.78; 122.05; 122.37; 136,57; 139.01; 149.47 (CH$_{aro}$); 157.49; 160.11; 164.03 (C$_{aro}$). HR-MS (Q-TOF): $m/z$, 230.1290; Calcd: 230.1293 for $[M + H]^+$.

HL$^{CHO}$ (3-*tert*-Butyl-4-hydroxy-5-{[(6-methoxy-pyridin-2-ylmethyl)-pyridin-2-ylmethyl-amino]-methyl}-benzaldehyde). To a solution of 2-*tert*-butyl-4-hydroxybenzaldehyde (6.9 mmol) in ethanol (20 mL), 2 molar equivalents of paraformaldehyde and 1 equivalent of (6-methoxy-pyridin-2-ylmethyl)-pyridin-2-ylmethyl-amine diluted in ethanol (25 mL) were added. After 3 days under stirring at room temperature, the solvent was evaporated under vacuum and the reaction mixture was extracted with dichloromethane, washed with a saturated NaCl solution, and dried over Na$_2$SO$_4$. The solution was evaporated under vacuum, yielding a yellow-brown oil. The raw product was purified by column chromatography on silica gel (ethyl acetate/pentane (2/8) and 5% methanol to afford HL$^{CHO}$ in a 46% yield. NMR $^1$H (400 MHz, CDCl$_3$): δ (ppm) = 1.44 (s, 9H *t*-Bu); 3.85 (s, 2H); 3.99 (s, 2H); 4.02 (s, 2H); 4.03 (s, 3H, OCH$_3$); 6.62 (d, 1H, $^3J_{H-H}$ = 8.2 Hz, H$_7$ or H$_9$); 6.82 (d, 1H, $^3J_{H-H}$ = 7.1 Hz, H$_7$ or H$_9$); 7.28 (m, 1H, H$_4$); 7.49 (m, 3H, H$_2$-H$_6$-H$_8$); 7.72 (d, 1H, $^4J_{H-H}$ = 1.9 Hz, H$_1$); 7.75 (m, 1H, H$_5$); 8.56 (dd, 1H, $^3J_{H-H}$ = 5 Hz, $^4J_{H-H}$ = 0.85 Hz, H$_3$); 9.79 (s, 1H, CHO). NMR $^{13}$C (Q.DEPT, 400 MHz, CDCl$_3$): δ (ppm) = 29.27 (3 CH$_3$, *t*-Bu); 34.91

(C, *t*-Bu); 53.51 (OCH$_3$); 57.46; 58.34; 59.09 (3 CH$_2$); 109.99; 116.22 (CH$_{aro}$); 122.79 (C$_{aro}$); 122.93; 124.14 (CH$_{aro}$); 127.96 (C$_{aro}$); 129.19; 129.92 (CH$_{aro}$); 137.65 (C$_{aro}$); 139.05 (CH$_{aro}$); 162.94; 164.18 (C$_{aro}$); 191.18 (CHO). MS (ESI): *m/z*, 420.3 [M + H]$^+$. Anal. Calcd for C$_{25}$H$_{29}$N$_3$O$_3$% C, 71.58; H, 6.97; N, 10.02. Found: C, 71.32; H, 6.84; N, 10.21%. HR-MS (Q-TOF): *m/z*, 420.2294; Calcd: 420.2287 for [M + H]$^+$.

HL$^{pyr}$ ((2-*tert*-Butyl-6-{[(6-methoxy-pyridin-2-ylmethyl)-pyridin-2-ylmethyl-amino]-methyl}-4-({4-[(pyren-1-ylmethyl)-amino]-butylamino}-methyl)-phenol). N$^1$-(pyren-1-ylmethyl)butane-1,4-diamine (400 mg, 0.48 mmol) and 3-*tert*-Butyl-4-hydroxy-5-{[(6-methoxy-pyridin-2-ylmethyl)-pyridin-2-ylmethyl-amino]-methyl}-benzaldehyde (145 mg, 0.48 mmol) were stirred overnight at room temperature in a MeOH/THF (3/1) mixture. Next, NaBH$_4$ (54 mg, 1.44 mmol) was added in small portions and the solution was stirred for two extra hours at room temperature. The reaction mixture was then extracted with AcOEt, washed with water and brine, and dried over anhydrous Na$_2$SO$_4$. The solution was evaporated under vacuum to afford yellow-brown oil. Column chromatography on silica gel with the eluent CH$_2$Cl$_2$/MeOH/Et$_3$N (92/5/3) afforded HL$^{pyr}$ as a white solid in a 75% yield. NMR $^1$H (400 MHz, CDCl$_3$): δ (ppm) = 1.37 (s, 9H, *t*-Bu); 1.64 (m, 4H, g and f); 2.64 (t, 2H, e or h); 2.85 (t, 2H, e or h); 3.57 (s, 2H, d); 3.59 (s, 2H, a); 3.63 (s, 2H, b); 3.71 (s, 2H, c); 4.00 (s, 3H, OCH$_3$); 4.50 (s, 2H, i); 6.57 (d, 1H, $^3J_{H-H}$ = 8.3 Hz, H$_7$); 6.72 (d, 1H, $^3J_{H-H}$ = 7.2 Hz, H$_9$); 7.01 (d, 1H, $^4J_{H-H}$ = 1.6 Hz, H$_2$); 7.11 (ddd, 1H, $^3J_{H-H}$ = 7.5 Hz; $^3J_{H-H}$ = 5.4 Hz; $^4J_{H-H}$ = 0.7 Hz, H$_4$); 7.42 (m, 2H, H$_6$ and H$_8$); 7.59 (td, 1H, $^3J_{H-H}$ = 7.5 Hz, $^4J_{H-H}$ = 1.7 Hz, H$_5$); 7.99–8.20 (m, 9H, 8H$_{pyrene}$ and H$_1$); 8.36 (d, 1H pyrene); 8,50 (d, 1H, $^3J_{H-H}$ = 5.4 Hz, H$_3$). NMR $^{13}$C (Q.DEPT, 400 MHz, CDCl$_3$): δ (ppm) = 27.82; 28.33 (2 CH$_2$ f and g); 29.61 (3 CH$_3$ *t*-Bu); 34.79 (C, *t*-Bu); 48.89; 49.93; 51.81; 53.55 (4 CH$_2$); 53.58 (OCH$_3$); 57.88; 59.12; 59.44 (3 CH$_2$); 109.67; 116.17; 122.30 (CH$_{aro}$); 122.71 (C$_{aro}$); 123.19; 123.61; 125.00 (CH$_{aro}$); 125.09 (C$_{aro}$); 125.28; 125.37; 126.12; 126.69; 127.35; 127.41; 127.53; 127.64; 128.07 (CH$_{aro}$); 129.24; 130.95; 131.00; 131.50; 133.67 (C$_{aro}$); 136.57 (CH$_{aro}$); 136.59 (C$_{aro}$); 138.94; 149.08 (CH$_{aro}$); 155.28; 155.83; 158.21; 164.11 (C$_{aro}$). HR-MS (Q-TOF): *m/z*, 706.4128; Calcd: 706.4121 for [M + H]$^+$.

HL$^{boc}$. The ligand (3-*tert*-Butyl-4-hydroxy-5-{[(6-methoxy-pyridin-2-ylmethyl)-pyridin-2-ylmethyl-amino]-methyl}-benzaldehyde) (300 mg, 0.74 mmol) and *tert*-butyl-N-(4-aminobutyl)carbamate (140 mg, 0,74 mmol) were solubilized in MeOH (30 mL). The solution was stirred 12 h at room temperature. After that time, sodium borohydride (84 mg, 2.22 mmol) was added over 3 h by small fractions. The reaction mixture was further stirred for 2 h and next extracted with AcOEt, washed with water and brine, and finally dried over anhydrous Na$_2$SO$_4$. The organic phase was evaporated under vacuum, affording the raw product as a yellow-brown oil. Purification by column chromatography on silica gel with the eluent CH$_2$Cl$_2$/MeOH (95/5) using a gradient of triethylamine (0–5%) as the eluent afforded HL$^{boc}$ as a colorless oil in a 64% yield. NMR $^1$H (400 MHz, CDCl$_3$): δ (ppm) = 1.41 (s, 9H, *t*-Bu); 1.42 (s, 9H, *t*-Bu); 1.53 (m, 4H, f and g); 2.65 (t, 2H, e); 3.10 (m, 2H, h); 3.67 (s, 2H, b); 3.74 (s, 2H, d); 3.83 (s, 2H, a); 3.84 (s, 2H, c); 4.02 (s, 3H, OCH$_3$); 6.59 (d, 1H, $^3J_{H-H}$ = 8.2 Hz, H$_7$); 6.80 (d, 1H, $^3J_{H-H}$ = 7.2 Hz, H$_9$); 6.88 (d, 1H, $^4J_{H-H}$ = 1.8 Hz, H$_1$ or H$_2$); 7.09 (d, 1H, $^4J_{H-H}$ = 1.8 Hz, H$_1$ or H$_2$); 7.14 (ddd, 1H, $^3J_{H-H}$ = 7.5 Hz, $^3J_{H-H}$ = 5.0 Hz, $^4J_{H-H}$ = 0.7 Hz, H$_4$); 7.47 (m, 2H, H$_6$ and H$_8$); 7.63 (td, 1H, $^3J_{H-H}$ = 7.5 Hz, $^4J_{H-H}$ = 1.8 Hz, H$_5$); 8.52 (d, 1H, $^3J_{H-H}$ = 5.0 Hz, H$_3$); 10.83 (s, 1H, OH phenol). NMR $^{13}$C (Q.DEPT, 400 MHz, CDCl$_3$): δ (ppm) = 27.13; 28.02 (2 CH$_2$, f and g); 28.59; 29.66 (6 CH$_3$, *t*-Bu); 34.90; 40.56 (2 C, *t*-Bu); 46.26; 48.93 (2 CH$_2$); 53.58 (OCH$_3$); 53.89; 58.19; 59.27; 59.60 (4 CH$_2$); 109.69; 116.26; 122.33 (CH$_{aro}$); 122.66 (C$_{aro}$); 123.68; 126.51; 127.63 (CH$_{aro}$); 128.98 (C$_{aro}$); 136.63; 138.95; 149.07 (CH$_{aro}$); 155.32; 155.79; 156.19; 158.24; 164.17 (C$_{aro}$). HR-MS (Q-TOF): *m/z*, 592.3873; Calcd: 592.3863 for [M + H]$^+$.

HL$^{amine}$. HL$^{boc}$ (202 mg, mg, 0.35 mmol) was solubilized in an ethanol solution (20 mL), which was saturated with gaseous HCl beforehand. The solution was stirred during 1 h at room temperature. The solvent was next evaporated under vacuum, quantitatively affording HL$^{amine}$ as a white solid. NMR $^1$H (400 MHz, D$_2$O): δ (ppm) = 1.30 (s, 9H, *t*-Bu); 1.78 (m, 4H, f and g); 3.06 (m, 4H, e and h); 4.13 (s, 5H, a and OCH$_3$); 4.18 (s, 2H, b);

4.33 (s, 2H, d); 4.46 (s, 2H, c); 7.13 (d, 1H, $^3J_{H-H}$ = 8.3 Hz, $H_7$); 7.23 (m, 3H, $H_1$-$H_2$-$H_9$); 7.83 (m, 2H, $H_4$-$H_6$); 8.06 (t, 1H, $^3J_{H-H}$ = 8,3 Hz, $H_8$); 8.33 (t, 1H, $^3J_{H-H}$ = 7.9 Hz, $H_5$); 8.66 (d, 1H, $^3J_{H-H}$ = 5.6 Hz, $H_3$). NMR $^{13}$C (Q.DEPT, 400 MHz, $D_2O$): δ (ppm) = 22.71; 23.99 (2 $CH_2$, f and g); 28.92 (3 $CH_3$, *t*-Bu); 34.08 (C, *t*-Bu); 38.81; 46.12; 50.48 (3 $CH_2$, a-e-h); 56.40 ($OCH_3$); 57.45; 57.82; 58.57 (3 $CH_2$, b-c-d); 109.56; 118.15 ($CH_{aro}$); 122.68; 123.56 ($C_{aro}$); 125.98; 127.07; 129.66; 130.42 ($CH_{aro}$); 139.07 ($C_{aro}$); 142.72; 145.20; 145.30 ($CH_{aro}$); 149.74; 151.22; 154.81; 162.63 ($C_{aro}$). HR-MS (Q-TOF): *m/z*, 492.3341; Calcd: 492.3339 for $[M + H]^+$.

$H_2L^{bis}$. (3-*tert*-Butyl-4-hydroxy-5-{[[(6-methoxy-pyridin-2-ylmethyl)-pyridin-2-ylmethyl-amino]-methyl}-benzaldehyde) (500 mg, 1.24 mmol) and 1,4-butane diamine (55 mg, 0.62 mmol) were solubilized in MeOH (30 mL). After 10 min stirring, the reductant $NaBH_4$ (6 molar equivalents, 141 mg, 3.72 mmol) was progressively added to the solution (over 3 h) at room temperature. After 2 h of additional stirring, the solvent was evaporated under vacuum and the reaction mixture was extracted with AcOEt, washed with water and brine, and dried over anhydrous $Na_2SO_4$. The organic phase was evaporated under vacuum, giving a yellow-brown oil. The product was purified by column chromatography on silica gel with $CH_2Cl_2$/MeOH (95/5) with a gradient of triethylamine (0–5%) as the eluent, affording $H_2L^{bis}$ as a white powder in a 50% yield. NMR $^1$H (400 MHz, $CDCl_3$): δ (ppm) = 1.39 (s, 18H, 6 $CH_3$, *t*-Bu); 1.65 (m, 4H, f); 2.66 (m, 4H, e); 3.67 (s, 4H, b); 3.72 (s, 4H, d); 3.80 (s, 4H, a); 3.82 (s, 4H, c); 4.01 (s, 6H, 2 $OCH_3$); 6.58 (d, 2H, $^3J_{H-H}$ = 8.2 Hz, $H_9$); 6.78 (d, 2H, $^3J_{H-H}$ = 7,2 Hz, $H_7$); 6.87 (d, 2H, $^4J_{H-H}$ = 1.6 Hz, $H_2$); 7.04 (d, 2H, $^4J_{H-H}$ = 1.6 Hz, $H_1$); 7.13 (dd, 2H, $^3J_{H-H}$ = 4.9 Hz, $^3J_{H-H}$ = 7.4 Hz, $H_4$); 7.44 (m, 4H, $H_6$ et $H_8$); 7.61 (td, 2H, $^3J_{H-H}$ = 7.4 Hz, $^4J_{H-H}$ = 1.6 Hz, $H_5$); 8.50 (d, 2H, $^3J_{H-H}$ = 4.9 Hz, $H_3$); 10.94 (s, 2H, OHphenol). NMR $^{13}$C (Q.DEPT, 400 MHz, $CDCl_3$): δ (ppm) = 27.70 ($CH_2$, f); 29.65 ($CH_3$, *t*-Bu); 34.86 (C, *t*-Bu); 48.35; 52.98 ($CH_2$); 53.59 ($OCH_3$); 58.01; 59.20; 59.42 ($CH_2$); 109.74; 116.30; 122,38 ($CH_{aro}$); 122.92 ($C_{aro}$); 123.69; 126.79; 128.06; 136.67 ($CH_{aro}$); 136.86; 137.73 ($C_{aro}$); 139,03; 149.12 ($CH_{aro}$); 155.26; 156.22; 158.11; 164.16 ($C_{aro}$). HR-MS (Q-TOF): *m/z*, 895.5621; Calcd: 895.5598 for $[M + H]^+$.

Copper complex **1**. The ligand $HL^{CHO}$ (40 mg, 0.095 mmol) was dissolved in methanol (1 mL). The salt $CuCl_2 \bullet 2H_2O$ (16.2 mg, 0.095 mmol) was dissolved in methanol (1 mL) and this solution was added under stirring to the solution of the ligand. Triethylamine was next added (14 mL, 0.1 mmol) and the solution was concentrated under vacuum. The crude product was dissolved in acetonitrile (2 mL) and methyl acetate was slowly diffused into this solution, affording single crystals of complex 1 (39 mg, yield 80%). HR-MS (Q-TOF): *m/z*, 481.1410; Calcd *m/z*: 481.1421 for $C_{25}H_{28}O_3N_3Cu$ ($[M]^+$). UV/Vis ($H_2O$:DMF 90:10, [NaCl] = 0.1 M, pH 7): 482 nm (ε = 662 $M^{-1}$ $cm^{-1}$). EPR (X-band, $H_2O$:DMF 90:10, [Tris−HCl] = 0.05 M, [NaCl] = 0.02 M, pH = 7, 100 K): $g_{\perp}$ = 2.056, $g_{//}$ = 2.257, $A_{\perp}$ = 1.4 mT, $A_{//}$ = 16.3 mT (with Cu).

Other complexes. Complex **2** and **3** were generated in situ by reacting equimolar amounts of $CuCl_2 \bullet 2H_2O$, base (triethylamine) and the appropriate ligand ($HL^{pyr}$, $HL^{amine}$, respectively) in DMF. Complex **4** was prepared in a similar way with $H_2L^{bis}$, except instead two molar equivalents of copper were used. The concentrated DMF solutions of the complexes were diluted in water in order to obtain a final medium of composition (DMF:$H_2O$) (10:90).

Complex **2**. HR-MS (Q-TOF): *m/z*, 384.1660; Calcd *m/z*: 384.1664 for $C_{46}H_{51}O_2N_5Cu$ ($[M + H]^{2+}$). UV/Vis ($H_2O$:DMF 90:10, [NaCl] = 0.1 M, pH 7): 462 nm (ε = 1040 $M^{-1}$ $cm^{-1}$). EPR (X-band, $H_2O$:DMF 90:10, [Tris−HCl] = 0.05 M, [NaCl] = 0.02 M, 100 K): $g_{\perp}$ = 2.054, $g_{//}$ = 2.254, $A_{\perp}$ = 1.5 mT, $A_{//}$ = 16.6 mT ($A_{Cu}$).

Complex **3**. HR-MS (Q-TOF): *m/z*, 277.1268; Calcd *m/z*: 277.1273 for $C_{29}H_{41}O_2N_5Cu$ ($[M + H]^{2+}$). UV/Vis ($H_2O$:DMF 90:10, [NaCl] = 0.1 M, pH 7): 461 nm (ε = 780 $M^{-1}$ $cm^{-1}$). EPR (X-band, $H_2O$:DMF 90:10, [Tris−HCl] = 0.05 M, [NaCl] = 0.02 M, 100 K): $g_{\perp}$ = 2.054, $g_{//}$ = 2.254, $A_{\perp}$ = 1.5 mT, $A_{//}$ = 16.6 mT ($A_{Cu}$).

Complex **4**. HR-MS (ESI-MS): *m/z*, 1020.45; Calcd *m/z*: 1020.41 for $C_{54}H_{70}Cu_2N_8O_4$ $[M + 2H]^+$. UV/Vis ($H_2O$:DMF 90:10, [NaCl] = 0.1 M, pH 7): 457 nm (ε = 2005 $M^{-1}$ $cm^{-1}$).

EPR (X-band, $H_2O$:DMF 90:10, [Tris$-$HCl] = 0.05 M, [NaCl] = 0.02 M, 100 K): $g_{\perp}$ = 2.054, $g_{//}$ = 2.254, $A_{\perp}$ = 1.5 mT, $A_{//}$ = 16.6 mT ($A_{Cu}$).

### 3.3. Crystal Structure Analysis

A single crystal of **1** was coated with a parafin mixture, picked up with nylon loops, and mounted in the nitrogen cold stream of the diffractometer. Mo-K$\alpha$ radiation ($\lambda$ = 0.71073Å) from a Mo-target X-ray micro-source equipped with INCOATEC Quazar mirror optics was used (INCOATEC, Geesthacht, Germany). Final cell constants were obtained from least squares fits of several thousand strong reflections. Intensity data were corrected for absorption using intensities of redundant reflections with the program SADABS [57]. The structures were readily solved by the charge flipping method. The OLEX2 1.2 software was used for the refinement [58]. All non-hydrogen atoms were anisotropically refined and hydrogen atoms were placed at calculated positions and refined as riding atoms with isotropic displacement parameters. CCDC-1479462 contains the crystallographic data for **1**; these data can be obtained free of charge via http://www.ccdc.cam.ac.uk/conts/retrieving.html (accessed on 25 September 2023).

### 3.4. Determination of the DNA Binding Constant

The binding constant towards CT-DNA was determined by fluorescence. CT-DNA (type I, fibers, from Sigma Aldrich) was purified beforehand, as reported [44]. Complex **2** (1 mM) was mixed with variable amounts of CT-DNA (typically 5–200 mM in base pairs) in a saline ($H_2O$:DMF) (90:10) solution ([NaCl] = 0.02 M), whose pH is buffered (pH = 7) by Tris$-$HCl (0.05 M). The fluorescence spectra were recorded after 10 min at 298 K. The DNA binding constant K was calculated by using a Scatchard–Von Hippel model from a duplicate experiment [54]: The ratio r/c was plotted against r, giving a straight line whose slope is K, where r represents the number of bound molecules per site and c corresponds to the concentration of free drug. The concentration of bound molecules was determined from $C_0 \times (f_0 - f)/(f_0 - f_b)$, where $f_0$ is the fluorescence of the free drug, f the fluorescence at any DNA concentration, $f_b$ the fluorescence of the drug bound to DNA, and $C_0$ the total concentration of drug (calculated from the mass balance).

### 3.5. Procedure for DNA Cleavage Experiments

The plasmidic DNA $\phi$X174 RF1 was purchased from Fermentas. The experiments were performed in a (water:DMF) (90:10174) mixture containing 10 mM phosphate buffer (pH = 7.2). In a typical experiment, double-stranded supercoiled DNA was incubated with the complexes at 37 °C during the appropriate time. The reaction was next quenched by decreasing the temperature to −20 °C. A loading buffer was added (6× loading dye solution, Fermentas) and the solution was loaded on a 0.8% agarose gel in Tris-Boric acid-EDTA buffer, (pH = 8) (0.5 × TBE). The electrophoresis was performed at 70 V during about 3 h. Once DNA has migrated, the gels were stained by a 10 min incubation with an ethidium bromide solution (1 mg/mL) followed by washing with distilled water. The images of the gels were recorded by using the imager Typhoon 9400 (Cytiva France, Saint Germain en Laye, France). The fluorescence was quantified using the IQ Solutions v1.4 Software.

### 3.6. Cell Culture

Human bladder cancer cell line RT112 was obtained from Cell Lines Service (Eppelheim, Germany). Cisplatin-resistant RT112 cells (RT112-CP) were kindly provided by B. Köberle (Institute of Toxicology, Clinical Centre of University of Mainz, Mainz, Germany). RT112 and RT112-CP cells were cultured in RPMI 1640 medium supplemented with 10% (*v/v*) fetal calf serum (FCS) and 2 mM glutamine (Invitrogen Life Technologies, Paisley, UK). Cells were maintained at 37 °C in a 5% $CO_2$-humidified atmosphere and tested to ensure freedom from mycoplasma contamination.

### 3.7. Cell Proliferation Assay

Inhibition of cell proliferation by copper complexes was measured by MTT (3-(4,5-Dimethylthiazol-2-yl)-2,5-diphenyltetrazolium bromide) assays. RT112 and RT112-CP cells were seeded into 96-well plates ($5 \times 10^3$ cells/well) in 100 µL of culture medium. After 24 h, cells were treated with cisplatin (Sigma-Aldrich, Lyon, France) or complexes at various concentrations. In parallel, a control with DMF (vehicle alone) at the same dilutions was performed. Following incubation for 48 h, 10 µL of MTT (Euromedex, Mundolsheim, France) stock solution in PBS at 5 mg/ml was added in each well and the plates were incubated at 37 °C for 3 h. Plates were then centrifuged 5 min at 1500 rpm before the medium was discarded and replaced with DMF (100 µL/well) to solubilize water-insoluble purple formazan crystals. After 15 min under shaking, absorbance was measured on an ELISA reader (Tecan, Männedorf, Switzerland) at a test wavelength of 570 nm and a reference wavelength of 650 nm. Absorbance obtained by cells treated with the same dilution of the vehicle alone (DMSO) was rated as 100% of cell survival. Each data point is the average triplicates of three independent experiments.

### 3.8. Computational Details

Full geometry optimizations were performed with the Gaussian 9 program [59], by using the B3LYP [60,61] functional. The 6–31g* basis set [62] was used for all the atoms. Frequency calculations were systematically performed in order to ensure that the optimized structure corresponds to a minimum and not a saddle point. Optical properties were computed by using time-dependent DFT (TD-DFT) [63], with the same basis set as for optimization. The solvent was taken into account by using a polarized continuum model (PCM) [64]. The 30 lowest energy excited states were calculated.

## 4. Conclusions

In summary, we prepared a series of copper tripodal complexes based on recently reported nucleases [44], in which the crucial $\alpha$-methylpyridine moiety is replaced by an $\alpha$-methoxypyridine. The methoxy group is both a stronger donor, and less sterically crowded. We establish both by X-Ray diffraction and DFT calculations that the copper ion geometry is significantly impacted by this substitution. In addition, the phenols' pKa are higher than for the "methyl" series [44], indicating weaker Lewis acidity at the metal center. Surprisingly, the Cu(II) reduction potential remains mostly unaffected ($E_p^{c,red} = -0.45$ to $-0.51$ V). The DNA cleavage activity of the complexes was investigated: Without reductant, all the compounds featuring a putrescine chain promote DNA condensation, which hampers direct observation of strand cleavage. On the other hand, **1** cleaves 20% of DNA at the highest concentration investigated (50 µM), whereas its methyl congener did not promote any cleavage at this concentration [44]. This result cannot be explained in terms of Lewis acidity of the metal center, since an opposite trend would be expected. It may instead reflect structural effects, whereby the metal is less sterically crowded in the present series, and thus is more accessible for generating nucleophiles.

When ascorbic acid is present the nuclease activity is significantly enhanced, and switches to an oxidative pathway (through OH$^\bullet$ formation). The IC$_{50}$ are 30, 7, 16, and >18 µM for complexes, **1**, **2**, **3**, and **4**, respectively. These values are in a narrow range, like the methyl derivatives (45, 1.7, 14 µM for the methyl derivatives of **1**, **2**, and **3**, respectively), with, however, two important differences: The activity of complex **1** is higher than that of the methyl congener, whereas that of complex **2** is lower. Thus, correlations between cleavage activity and nature of the pyridine $\alpha$-substituent depend on more than a single parameter. The methoxy substitution herein appears more interesting for complexes that are not vectorized towards phosphate by the putrescine chain. Notably, the best nuclease in this series is complex **2** under reducing conditions. Although slightly less efficient than its methyl congener [44], it remains a very efficient nuclease. Its lower activity may be attributed to the slightly lower binding constant towards DNA ($1.6 \times 10^6$ M$^{-1}$ vs $2.6 \times 10^6$ M$^{-1}$). Complexes **2** and **4**, which are the best nucleases, inhibit

the proliferation of bladder cancer cells much more efficiently than cisplatin (5 to 25-times better). The IC$_{50}$ of complex **4** is slightly smaller than that of complex **2** (1–1.6 µM), but if the concentration is calculated on the basis of the copper content, complex **2** is the best agent. Finally, both overcome the resistance to cisplatin of RT112-CP cells.

In summary, the results on DNA cleavage demonstrate that a hydrolytic pathway can be favored when the steric bulk in α-position of the pyridine is decreased (methoxy vs. methyl substituent) [44]. The oxidative pathway is unfavored when a methoxy substituent replaces the methyl one, which is attributed to electronic rather than steric effects. These trends provide important insight into the strategy to be used for further functionalization and bioconjugation of the ligands, especially the incorporation of poly(O-alkyl) chains (PEG) instead of simple O-methyl groups, with the aim of enhancing the delivery of the copper nuclease once injected in an organism [65].

**Supplementary Materials:** The following supporting information can be downloaded at: https://www.mdpi.com/article/10.3390/inorganics11100396/s1, Figures S1–S4: $^1$H NMR spectra of the ligands; Figures S5–S8: HR-MS of the ligands; Figure S9: Mass spectra of the complexes; Figure S10: Jobs' plot of **4**; Figures S11–S15: EPR spectra of the complexes; Figures S16–S18: Electronic spectra of the complexes; Figures S19–S24: CV curves of the complexes; Figures S25–S26: Agarose gel electrophoresis; Figure S27: summarizes the atom numbering used for assigning the $^1$H NMR resonances. Output of geometry optimizations and optical properties.

**Author Contributions:** Conceptualization, F.T. and X.R.; funding acquisition, F.T.; investigation, D.S., S.E. and C.P.; methodology, V.M.-F. and N.B.; project administration, F.T.; writing—review & editing, F.T., V.M.-F. and N.B. All authors have read and agreed to the published version of the manuscript.

**Funding:** This research was funded by the French National Research Agency in the framework of the "Investissements d'avenir" program (ANR-15-IDEX-02) and Labex ARCANE and CBH-EUR-GS (ANR-17-EURE-0003).

**Data Availability Statement:** The data are given in Supplementary Materials.

**Acknowledgments:** The NanoBio-ICMG platforms (UAR 2607) are acknowledged for their technical support. The authors thank the CECIC cluster for providing computational resources.

**Conflicts of Interest:** The authors declare no conflict of interest.

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
