# Peer review of "Anti-Proliferation and DNA Cleavage Activities of Copper(II) Complexes of N3O Tripodal Polyamine Ligands"

_inorganics, doi:10.3390/inorganics11100396_

Round 1

Reviewer 1 Report

This well-written manuscript reports on a series of newly prepared copper complexes and their DNA interaction, DNA cleavage as well as their anticancer activity. The work is sound. The manuscript deserves publication.

Author Response

Dear editor, dear reviewer,

We are glad to submit a revised version of our paper entitled “Anti-proliferation and nuclease activities of copper(II) complexes of tripodal polyamine ligands” by Doti Serre, Sule Erbek, Nathalie Berthet, Christian Philouze, Xavier Ronot, Véronique Martel-Frachet and myself as a full paper. We would like to thank all the reviewers and the editor for their assessment of the manuscript and their suggestions, which enabled us to improve its quality. We agree with all the comments and hope that this new version will meet all their expectations.

Specific answer to the reviewers’comments:

Comment: This well-written manuscript reports on a series of newly prepared copper complexes and their DNA interaction, DNA cleavage as well as their anticancer activity. The work is sound. The manuscript deserves publication.

Answer: We thank reviewer 1 for her/his positive evaluation of our manuscript.

Reviewer 2 Report

The manuscript is well-written and significantly contributes to the better understanding of the nuclease activity of copper(II) complexes. The experiments were well-planned and conclusions are sound.

One very minor comment:

The absorption spectra are shown in Figure 3 and in the supplement, but these Figures are two small to observe the differences. It would be useful to list the parameters of d-d transitions (lambda/epsylon) in a separate Table, at least in the Supplement.

Acceptable

Author Response

Dear editor, dear reviewer,

We are glad to submit a revised version of our paper entitled “Anti-proliferation and nuclease activities of copper(II) complexes of tripodal polyamine ligands” by Doti Serre, Sule Erbek, Nathalie Berthet, Christian Philouze, Xavier Ronot, Véronique Martel-Frachet and myself as a full paper. We would like to thank all the reviewers and the editor for their assessment of the manuscript and their suggestions, which enabled us to improve its quality. We agree with all the comments and hope that this new version will meet all their expectations.

Specific answer to the reviewers’comments:

The manuscript is well-written and significantly contributes to the better understanding of the nuclease activity of copper(II) complexes. The experiments were well-planned and conclusions are sound.

One very minor comment:

The absorption spectra are shown in Figure 3 and in the supplement, but these Figures are two small to observe the differences. It would be useful to list the parameters of d-d transitions (lambda/epsylon) in a separate Table, at least in the Supplement.

Answer: We thank reviewer 2 for her/his positive evaluation of our manuscript. Table 2 has been updated to include the parameters of the d-d transitions, while four new figures were included in ESI, showing only the spectra at low and high pHs.

Reviewer 3 Report

The present work was performed based on a previous one published in 2018 by the same group. The data reported is not enough to justify a publication, the improvements from the 2018 paper are essentially none. I do not recommend publication.

The quality of English is acceptable, anyhow grammar and spelling merit a recheck.

Author Response

Dear editor, dear reviewer,

We are glad to submit a revised version of our paper entitled “Anti-proliferation and nuclease activities of copper(II) complexes of tripodal polyamine ligands” by Doti Serre, Sule Erbek, Nathalie Berthet, Christian Philouze, Xavier Ronot, Véronique Martel-Frachet and myself as a full paper. We would like to thank all the reviewers and the editor for their assessment of the manuscript and their suggestions, which enabled us to improve its quality. We agree with all the comments and hope that this new version will meet all their expectations.

Specific answer to the reviewers’comments:

The present work was performed based on a previous one published in 2018 by the same group. The data reported is not enough to justify a publication, the improvements from the 2018 paper are essentially none. I do not recommend publication.

Answer: We agree that the there is not a large enhancement of the biological activity of the compounds in this new series. The aim of this paper was not necessarily the improvement of the biological activity, but rather the evaluation of modifications of the steric bulk and electronic effect of the substituent of the a-pyridine substituent. The methoxy substituent was specifically investigated since it is a simple model for PEG chains. These results demonstrate that sophisticated PEG derivatives could be synthesized without significant drop of nuclease activity. Furthermore the hydrolytic cleavage activity is enhanced for some compounds harboring the methoxy substituent, whereas the oxidative cleavage is decreased. Since trend offers new perspective for tuning the activity of this family of compounds.

We updated the conclusion by adding the following sentences :

In summary the results on DNA cleavage demonstrate that a hydrolytic pathway can be favoured when the steric bulk in a-position of the pyridine is decreased (methoxy vs. methyl substituent). The oxidative pathway is unfavoured when a methoxy substituent replaces the methyl one, which is attributed to electronic rather than steric effects. These trends provide important insight onto the strategy to be used for further functionalization and bioconjugation of the ligands, especially the incorporation of poly(O-alkyl) chains (PEG) instead of simple O-methyl groups, with the aim of enhancing the delivery of the copper nuclease once injected in an organism.[67]